# What vaccination rate(s) minimize total societal costs after 'opening up' to COVID-19? Age-structured SIRM results for the Delta variant in Australia (New South Wales, Victoria and Western Australia)

**Long Chu**[1], **R. Quentin Grafton**[1], **Tom Kompas**[1,2]*

**1** Crawford School of Public Policy, Australian National University, Canberra, Australia, **2** Centre of Excellence for Biosecurity Risk Analysis, School of Biosciences and School of Ecosystem and Forest Sciences, University of Melbourne, Melbourne, Australia

* tom.kompas@unimelb.edu.au

## Abstract

Using three age-structured, stochastic SIRM models, calibrated to Australian data post July 2021 with community transmission of the Delta variant, we projected possible public health outcomes (daily cases, hospitalisations, ICU beds, ventilators and fatalities) and economy costs for three states: New South Wales (NSW), Victoria (VIC) and Western Australia (WA). NSW and VIC have had on-going community transmission from July 2021 and were in 'lockdown' to suppress transmission. WA did not have on-going community transmission nor was it in lockdown at the model start date (October 11th 2021) but did maintain strict state border controls. We projected the public health outcomes and the economic costs of 'opening up' (relaxation of lockdowns in NSW and VIC or fully opening the state border for WA) at alternative vaccination rates (70%, 80% and 90%), compared peak patient demand for ICU beds and ventilators to staffed state-level bed capacity, and calculated a 'preferred' vaccination rate that minimizes societal costs and that varies by state. We found that the preferred vaccination rate for all states is at least 80% and that the preferred population vaccination rate is increasing with: (1) the effectiveness (infection, hospitalization and fatality) of the vaccine; (2) the lower is the daily lockdown cost; (3) the larger are the public health costs from COVID-19; (4) the higher is the rate of community transmission before opening up; and (5) the less effective are the public health measures after opening up.

## 1. Introduction

Most national governments wish to vaccinate their populations as fast as possible to reduce the likelihood of hospitalizations and fatalities from COVID-19 caused by the SARS-CoV-2 virus. For countries with widespread community transmission and public health measures in place intended to reduce transmission, a key public policy question is: What is the preferred

**Data Availability Statement:** All data and model source code is contained in the online repository: https://osf.io/zu3bn/.

**Funding:** The authors received no specific funding for this work.

**Competing interests:** The authors have declared that no competing interests exist.

vaccination rate(s) to trigger relaxation of public health measures ('opening up') that may include school closures, work-from-home advisories, temporary closure of most retail shops, etc.? From a total societal cost perspective this involves a trade-off. That is, a higher vaccination rate prior to opening up will, all else equal, reduce public health costs but the delay to achieve a higher vaccination rate may increase cumulative economy lockdown costs. We calculated these trade-offs to estimate a 'preferred' population vaccination rate for opening up an economy that has either stringent public health measures in place, that we labelled a 'lockdown', or strict border controls, such as supervised 14 days quarantine, for all arrivals.

We used data for Delta variant community transmission from July 1st 2021 and for three states in Australia: New South Wales (NSW), Victoria (VIC) and Western Australia (WA). We projected using separate state-level Susceptible-Infected-Recovered-Mortality (SIRM) models, the state-level public health outcomes associated with opening up after 70%, 80% and 90% of the eligible (12 years and older) state populations were fully vaccinated, from a model start date (October 11th 2021). At the model start date for our SIRM projections, NSW had widespread community transmission of the Delta variant but declining new daily cases, VIC had widespread community transmission of the Delta variant with increasing new daily cases, and WA had no community transmission.

The NSW and VIC epidemics began from an initial outbreak first identified in Sydney on June 16th 2021 [1]. In response to their state-wide epidemics, the NSW and VIC state governments imposed a variety of public health measures including school closures, work-from-home advisories, mandated mask wearing, among other measures. As the model start date, WA did not have community transmission, and only had minimal public health measures (no lockdown). Nevertheless, WA did have strict arrival protocols, including an entry permit for all arrivals, COVID-19 testing, and 14-days supervised quarantine for all international and domestic arrivals from Australian jurisdictions with community transmission.

Australia provides a valuable comparison to other countries when assessing the preferred population vaccination rate to begin relaxation of stringent public health measures to reduce COVID-19 community transmission. This is because of: (1) the diversity of public health approaches within Australia and across its state jurisdictions; (2) differences in community transmission across Australian states; (3) as of October 2021, a low rate of COVID-19 infection such that almost the entire population wass highly susceptible to COVID-19 in the absence of vaccination; (4) an agreed-to-National Plan in relation to opening up and the use of lockdowns based on 70% and 80% national vaccination levels of those aged 16 years and above [2]; (5) multiple state and/or national epidemiological models to compare projected health outcomes [3–7]; and (6) an *ex-ante* opportunity to determine the preferred vaccination rate by state.

Our contributions are three-fold. First, we developed separate state-based age-structured SIRM models for NSW, Victoria and WA calibrated to the Delta variant. Second, our projections from the SIRM models provide an opportunity to assess public health capacity (ICU beds and ventilators) to projected public health demand under multiple vaccination rate scenarios. Third, we used Australian Treasury estimates of lockdown costs [8], estimates of health care costs [9], including losses associated with those who recover, and estimates of the dollar loss of fatalities, adjusted for age with a Value of Statistical Life (VSL) used by the Australian government [10]. Using these costs and SIRM models we estimated the preferred vaccination rate that minimizes total societal costs from opening up. Our methods, where suitable data are available, could be applied to any jurisdiction, both *ex-ante* and *ex-post*, to determine a preferred population vaccination rate(s) to relax stringent public health measures to control COVID-19 transmission.

## 2. Materials and methods

### 2.1. Model Specification

Australia has six states and two territories. Each jurisdiction is responsible for its own public health measures and has the ability to control entry and exit to and from its jurisdictional borders. We separately estimated a separate age-structured SIRM for NSW, VIC and WA. Collectively, these three states represent more than two thirds of the Australian population and economy.

**2.1.1. Age cohorts and population.** Table 1 summarizes the population of three Australian states, NSW and VIC–the two most populous Australian states, and WA—the largest state by surface area. There are ten age groups in each state, following the age group classifications of the Australian Bureau of Statistics (ABS), for each of the three-age structured SIRM models separately estimated for each state.

Each group $i \in [1,2,\ldots,10]$ is further classified into two categories, unvaccinated and vaccinated. Each category has its own SIRM compartment. The two categories are linked by the flow of vaccination from the unvaccinated to the vaccinated category. Our model only considered fully vaccinated (with two doses) people labelled 'vaccinated' and those who have either no dose or only one dose labelled 'unvaccinated'.

**2.1.2. Multi-age-group infections.** Susceptible people in group $i$, either unvaccinated or vaccinated, may get infected during contact with infected people, i.e., the source of infection. The source of infection can be in any of the age groups, $j \in [1,2,..10]$, including the same group of the susceptible people (group $i$). The probability of a susceptible person in group $i$ getting infected depends on: (i) the relative susceptibility of group $i$ which is presented by parameter $\gamma_i$, (ii) whether the susceptible person (in group $i$) has been vaccinated–vaccination can reduce the susceptibility of people in age group $i$ by $\sigma_i \in (0,1)$, and (iii) the contagiousness of the source of infection in age group $j$ toward susceptible people in age group $i$—the relative contagiousness of unvaccinated source of infection (denoted as $\alpha_{ij}$) and the vaccine effectiveness with respect to onward transmission, i.e., how much vaccination can reduce onward transmission (denoted as $\beta_{ij}$).

The dynamics of each age group $i \in [1,2..10]$ are formalised in Eqs (1)–(8) following the SIRM diagram in Fig 1. The first four equations present the unvaccinated compartments for

**Table 1. Australian, NSW, VIC, and WA age cohorts (millions of people).**

| Group index (i) | Age | NSW | | VIC | | WA | |
|---|---|---|---|---|---|---|---|
| | | Total | 12+ | Total | 12+ | Total | 12+ |
| 1 | 0–9 | 1.01 | 0.00 | 0.82 | 0.00 | 0.35 | 0.00 |
| 2 | 10–19 | 0.97 | 0.77 | 0.77 | 0.61 | 0.33 | 0.26 |
| 3 | 20–29 | 1.15 | 1.15 | 1.01 | 1.01 | 0.35 | 0.35 |
| 4 | 30–39 | 1.19 | 1.19 | 1.03 | 1.03 | 0.4 | 0.4 |
| 5 | 40–49 | 1.04 | 1.04 | 0.86 | 0.86 | 0.35 | 0.35 |
| 6 | 50–59 | 0.98 | 0.98 | 0.79 | 0.79 | 0.33 | 0.33 |
| 7 | 60–69 | 0.87 | 0.87 | 0.68 | 0.68 | 0.27 | 0.27 |
| 8 | 70–79 | 0.61 | 0.61 | 0.47 | 0.47 | 0.18 | 0.18 |
| 9 | 80–89 | 0.28 | 0.28 | 0.22 | 0.22 | 0.08 | 0.08 |
| 10 | 90+ | 0.07 | 0.07 | 0.06 | 0.06 | 0.02 | 0.02 |
| Total | | 8.17 | 6.96 | 6.71 | 5.73 | 2.66 | 2.24 |

Notes: The population in each age group is extracted from ABS statistics in millions of people together with 12+ people who are eligible for Covid-19 vaccination, rounded to the nearest 2-decimal places.

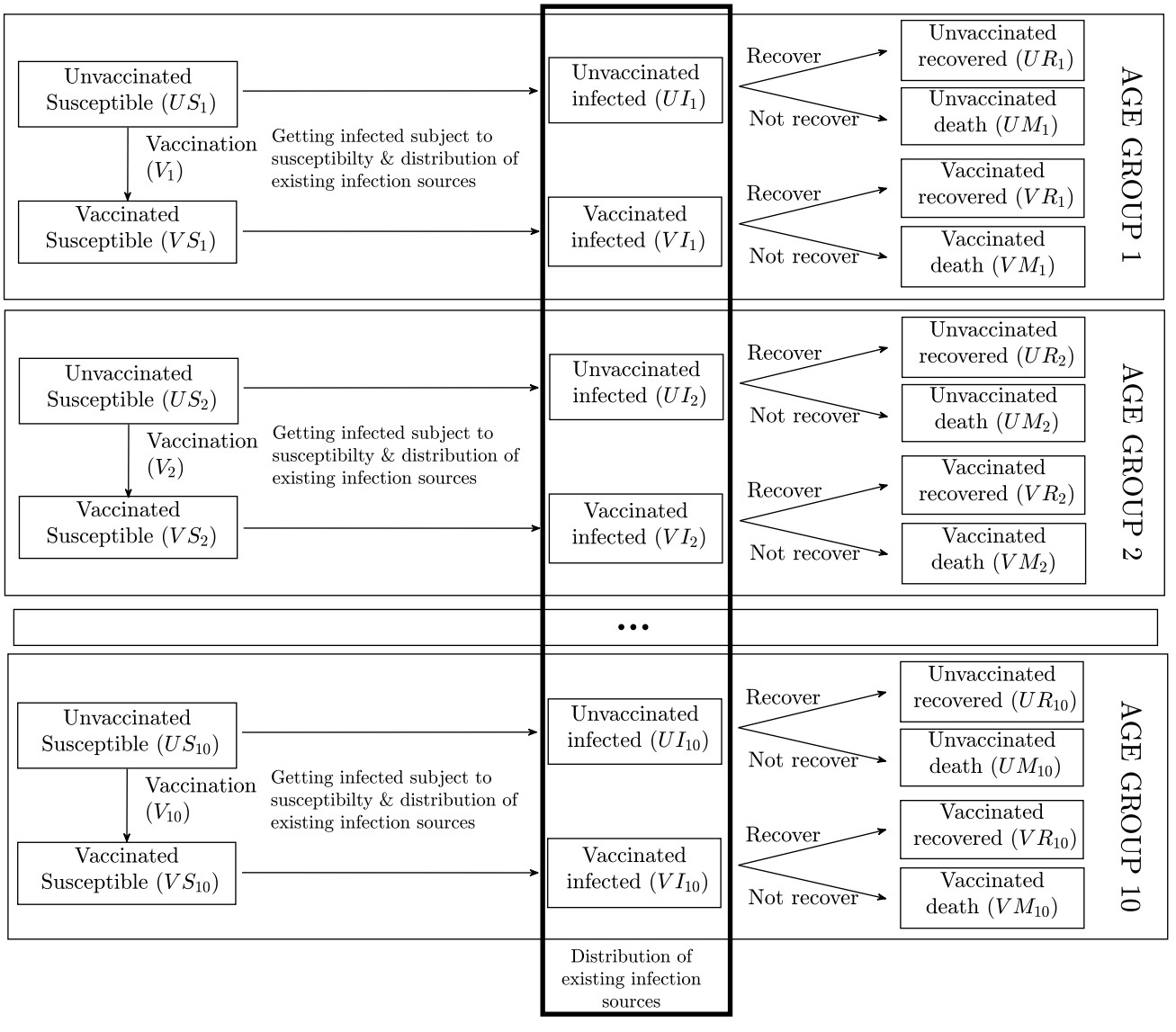

**Fig 1. Age-structured SIRM compartments.**

each age group, and the remaining four equations represent the vaccinated compartments noting we do not account for any monthly population growth.

The parameter $R_0$ is the average (unmitigated) basic reproduction rate, $N_i$ is the population of age group $i$, $T_i$ is the average unmitigated exposure time of an infection source, $\Gamma_i$ is the effectiveness of control measures (i.e., how effective are public health measures at reducing community transmission within each age cohort), $ru_i$ and $rv_i$ are the finalisation rates (either recovery or death) of unvaccinated and vaccinated patients in age group $i$. Further, $mu_i$ and $m_i$ are, respectively, the mortality rate of unvaccinated patients in age group $i$ and the vaccine effectiveness on mortality (i.e., how vaccination reduce mortality risk). Compartmental variables are defined in Fig 1.

$$\frac{\Delta US_i}{\Delta t} = -V_i - \frac{\gamma_i N_i R_0}{\sum_{j=1}^{10} \gamma_j N_j T_j}(1 - \Gamma_i)US_i \sum_{j=1}^{N}(\alpha_{ij}UI_j + (1 - \beta_{ij})\alpha_{ij}VI_j) \qquad (1)$$

$$\frac{\Delta UI_i}{\Delta t} = \frac{\gamma_i N_i R_0}{\sum_{j=1}^{10} \gamma_j N_j T_j}(1 - \Gamma_i)US_i \sum_{j=1}^{N}(\alpha_{ij}UI_j + (1 - \beta_{ij})\alpha_{ij}VI_j) - UI_i \, ru_i \qquad (2)$$

$$\frac{\Delta UR_i}{\Delta t} = UI_i \, ru_i(1 - mu_i) \qquad (3)$$

$$\frac{\Delta UM_i}{\Delta t} = UI_i \, ru_i mu_i \qquad (4)$$

$$\frac{\Delta VS_i}{\Delta t} = V_i - (1 - \sigma_i)\frac{\gamma_i N_i R_0}{\sum_{j=1}^{10} \gamma_j N_j T_j}(1 - \Gamma_i)VS_i \sum_{j=1}^{N}(\alpha_{ij}UI_j + (1 - \beta_{ij})\alpha_{ij} \, VI_j) \qquad (5)$$

$$\frac{\Delta VI_i}{\Delta t} = (1 - \sigma_i)\frac{\gamma_i N_i R_0}{\sum_{j=1}^{10} \gamma_j N_j T_j}(1 - \Gamma_i)VS_i \sum_{j=1}^{N}(\alpha_{ij}UI_j + (1 - \beta_{ij})\alpha_{ij} \, VI_j) - VI_i \, rv_i \qquad (6)$$

$$\frac{\Delta VR_i}{\Delta t} = VI_i \, rv_i(1 - mu_i(1 - m_i)) \qquad (7)$$

$$\frac{\Delta VM_i}{\Delta t} = VI_i \, rv_i mu_i(1 - m_i) \qquad (8)$$

**2.1.3. Hospitalizations, ICU admissions, and ventilation requirements.** We assumed the number of hospitalisations due to COVID-19 is a fixed proportion of the number of infected patients, where the fraction varies across age groups and whether the patients have been vaccinated. The estimated number of hospitalizations is formalized in Eq (9) for unvaccinated patients ($UH_i$) and in Eq (10) for vaccinated patients ($VH_i$). In these equations, $h_i$ and $vh_i$ are, respectively, the average hospitalization rate of unvaccinated cases and the vaccine effectiveness to avoid hospitalization (i.e., how much vaccination reduces the risk of hospitalization) of age group $i$. Similarly, the estimated ICU admissions are formalized in Eqs (11) and (12) where $icu_i$ and $vicu_i$ are the rate of ICU admissions for unvaccinated cases and the vaccine effectiveness on ICU admissions, respectively. The estimated ventilation requirements are formalized in Eqs (13) and (14) where $ven_i$ and $vven_i$ are the average rate of ventilation requirements for unvaccinated cases and the vaccine effectiveness to avoid ventilation requirements.

$$UH_i = UI_i \times h_i \qquad (9)$$

$$VH_i = VI_i \times h_i(1 - vh_i) \qquad (10)$$

$$UICU_i = UI_i \times icu_i \qquad (11)$$

$$VICU_i = VI_i \times icu_i(1 - vicu_i) \qquad (12)$$

$$UVen_i = UI_i \times ven_i \qquad (13)$$

$$VVen_i = VI_i \times ven_i(1 - vven_i) \qquad (14)$$

**2.1.4. Vaccination progress.**   We projected the daily vaccination progress in an age group as proportional to the number of (eligible) susceptible people who have not been fully vaccinated in that age group as specified in Eq (15). In this equatoin, *TV* is the total number of people becoming fully vaccinated in all 10 age groups on a particular day. Eq (15) implies that if an age group has fewer susceptible people who are eligible but not vaccinated, the vaccination progress of that age group, measured by the number of people, would be lower (or even zero if there are no unvaccinated susceptible people in that age group). We do not have data on the daily number of people who became fully vaccinated after recovering from COVID19, and we assume that this number, if any, is negligible.

$$V_i = TV \frac{US_i}{\sum_{j=1}^{10} US_j} \tag{15}$$

## 2.2. Data and parameterization

**2.2.1. Summary of key COVID19 data.**   Table 2 summarizes COVID-19 outcomes at the model start date, October 11[th] 2021, in Australia, and separately for NSW, VIC and WA. This table shows, at that date, that NSW and VIC accounted for more than 95% of the total COVID-19 cases. Epidemiological indicators vary across the states with substantial differences in terms of the hospitalization rates, the ICU admission rate, and the ventilation rate since the start of the pandemic.

The progress of vaccination has varied across the states. For example, about 58% of the eligible (12+) population in NSW were fully vaccinated as of the model start date (October 11[th] 2021), and the number of fully vaccinated people was increasing by about 67,000 per day in NSW over the preceding 14-day period. By comparison, on the model start date, about 46% of

**Table 2. Summary of key COVID-19 Statistics in NSW, VIC, and WA, as of October 11[th] 2021.**

|                                                              | NSW       | VIC       | WA        |
| ------------------------------------------------------------ | --------- | --------- | --------- |
| Total cases (1000 people)                                    | 69.21     | 54.47     | 1.11      |
| Active cases (1000 people)                                   | 6.75      | 19.01     | 0         |
| Hospital cases (1000 people)                                 | 769       | 677       | 0         |
| [Hospitalization rate]                                       | [11%]     | [3.7%]    |           |
| ICU cases (1000 people)                                      | 0.153     | 0.133     | 0         |
| [ICU admission rate]                                         | [2.2%]    | [0.7%]    |           |
| Cases on ventilators (1000 people)                           | 0.071     | 0.094     | 0         |
| [Ventilation requirement rate]                               | [1.1%]    | [0.5%]    | [NA]      |
| Recovered (1000 people)                                      | 61.96     | 34.546    | 1.1       |
| Total deaths (1000 people)                                   | 499       | 913       | 0.01      |
| [Mortality Rate]                                             | [0.8%]    | [0.6%]    | [NA]      |
| 14-day new-case trend with control measures (1000 people/day)| 0.72      | 1.616     | 0         |
| Fully vaccinated (million people)                            | 4.92      | 3.23      | 1.10      |
| [Full vaccination/eligible population]                       | [70.7%]   | [56.6%]   | [49.1%]   |
| 14-day vaccination rate (million people /day)                | 0.063     | 0.045     | 0.011     |

Notes:

1. Data sources are reported in S1 Text.

2. 14-day new case trend, the 14-day vaccination rates, the rates of hospitalisation, ICU admission, and ventilation are estimated using the average of the 14-day period ending October 11[th] 2021. The mortality rates only are estimated using data from July 1[st] 2021 that coincident with the Delta epidemic in NSW and VIC.

**Table 3. Age distribution of key epidemiological parameters.**

| Age groups | Relative susceptibility (Ordinal susceptibility units) | Hospitalisation rate (% of cases) | ICU rate (% of cases) | Mortality rate (% of cases) |
|---|---|---|---|---|
| 0–9 | 0.34 | 0.001 | 6E-05 | 4E-05 |
| 10–19 | 0.66 | 0.004 | 1E-04 | 4E-05 |
| 20–29 | 1.00 | 0.019 | 8E-04 | 2E-04 |
| 30–39 | 1.00 | 0.052 | 0.002 | 7E-04 |
| 40–49 | 1.00 | 0.074 | 0.004 | 0.002 |
| 50–59 | 1.00 | 0.171 | 0.019 | 0.006 |
| 60–69 | 1.24 | 0.283 | 0.071 | 0.016 |
| 70–79 | 1.47 | 0.413 | 0.122 | 0.049 |
| 80–89 | 1.47 | 0.449 | 0.031 | 0.158 |
| 90+ | 1.47 | 0.449 | 0.031 | 0.287 |

Source: [3: Table S3].

VIC's eligible population was fully vaccinated and it had a daily vaccination rate of 45,000 people over the preceding 14 days.

**2.2.2. Epidemiological parameters.** Epidemiological parameters vary significantly across age groups. Table 3 provides the age distribution of the epidemiological parameters from [3: Table S3]. For each jurisdiction (NSW, VIC and WA), we scaled the age distribution in Table 3 to match the aggregate values observed in actual data in Table 2. In addition, we assumed the age distribution of the ventilation requirement was similar to that of the ICU admissions.

COVID-19 is highly infectious if not mitigated. We assumed a daily transmission rate of COVID-19 equivalent to the reproduction number that varies between 4.0 and 8.0 [11], i.e., ($R_0 \sim [4,8]$). We also assumed that a low level of restrictions in NSW and VIC (social distancing, masks, hygiene, quarantine, and self-isolation of cases) reduced the transmission by 20%-40% ($\Gamma_i \sim [0.2, 0.4]$) as adapted from previous studies [9,12] and by 5%-15% ($\Gamma_i \sim [0.05, 0.15]$) in WA where minimal public health safety measures were in place because there was no community transmission in that state at the model start date.

Under the most recent lockdowns in 2021, community transmission in NSW and VIC was assumed to generate daily new cases between 80% and 120% of what has been observed during the 14-day period ending at the model start date, noting that WA had no community transmission at the model start date. The average recovery time for COVID-19 patients, if they can recover and do not suffer from 'long COVID' symptoms, is 14 days [13], i.e., $rv_i = uv_i = 365/14$.

**2.2.3. Vaccination targets, vaccination progress, and vaccine effectiveness.** For each jurisdiction, we considered three vaccination targets for removing lockdown measures, namely 70%, 80% and 90% of the eligible population (12 years and older) are fully vaccinated. The vaccination progress was assumed to be similar to the average vaccination rate during the 14-day period ending at the model start date of October 11th 2021 and until 90% of the eligible (12 years and older) population was fully vaccinated in the respective states of NSW, VIC and WA. In all cases, we assumed that opening up was irreversible and, thus, we do not consider lockdowns or any subsequent re-introduction of lockdowns [14]. We acknowledge the possibility that the number of cases might possibly grow past 10% of the eligible population while lockdown is maintained to achieve the 90% vaccination rate (and less than 90% of population were susceptible–assuming no reinfection). In the case of Australia, this is unlikely outcome given the relatively small number of cases (around 0.5% population), and a vaccination rate of 50% or greater on October 11th 2021, and the relatively rapid vaccination program.

Two main vaccines used in Australia are Astra Zeneca and Pfizer with different intervals between 2 doses, 12 and 3 weeks respectively. Initially, Australia prioritized the elderly for vaccination as they are more vulnerable to COVID19 than younger aged cohorts which were allowed to be vaccinated only after vaccines became more available. On day zero of our model (October 11[th] 2021), all eligible people (12+) had access to vaccines if they wished. Due to the lack of publicly available daily data on the age distribution of Astra Zeneca and Pfizer rollout, it was not possible for us to project how many people in each age cohort become fully vaccinated on a given day. Before the the 90% vaccination rate target is reached, we projected the total number of people becoming fully vaccinated (2 doses) on a given day by Eq (15) and for it to be similar to the 14-day average preceding day zero, and then varied this total number in a sensitivity analysis.

We assumed that ffter the 90% vaccination rate target was reached, the vaccination rate slows, and this total number was projected by 1% of the remaining unvaccinated susceptible people, a decay rate adapted from the Australian Capital Territory (ACT), a jurisdiction surrounded by NSW and which had the most rapid rate of vaccination and has the highest vaccination rate (98.5% of persons aged 12 years and older have received 2 doses) in Australia. The projected slow down in the vaccination rate by state is consistent with a flattening of the speed of vaccination at high vaccination rates [15]. We also specified that the proportion of the Australian population (and for each state for each age cohort) that is vaccine resistant such that they would 'Definetly not' get vaccinated is 5% [16]. We note that the proportion of those unwilling to be vaccinated in Australia declined from 9.4% to 8.2% in September 2021 [17].

In relation to vaccine effectiveness, we used the parameters published by [4: Table S2.3 and S2.5] for Australia, as given in Table 4. When there were significant differences across vaccines in relation to effectiveness, we calculated their average levels ($\sigma_i = 0.7$, $\beta_{ij} = 0.65$, $vh_i = 0.87$, $vicu_i = 0.88$, $vven_i = 0.88$, $m_i = 0.9$). We assumed that the vaccine effectiveness remains stable, with booster doses when necessary.

**2.2.4. Health care, welfare, and economy costs.** Parameters for the health care and welfare costs are from [9]. The cost of a standard hospital bed was 1,839 AUD/day, and the cost of an ICU bed was 4,885 AUD/day. The value of a statistical life year (VSLY) was 217,000AUD [10]. If a COVID-19 patient can recover without long-COVID symptoms, we assumed the welfare loss of the patient was equivalent to the value of time required for recovery as a proportion of the VSLY. We assumed that 10% of patients have long COVID symptoms, i.e., they are no longer infectious, but some of their symptoms persist for up to 12 weeks or longer.

We specified that the VSL was 5.0 million AUD [10] for people below 60 years of age. For those 60 years and older, we calculated an adjusted-VSL that was assumed to decrease linearly to 10% of the VSL (more than twice of the estimated VSLY) until 89-years of age. For 90-year-olds and above, the adjusted-VSL was fixed at 10% of the unadjusted VSL, or 500,000 AUD.

Our measures of economy costs of lockdowns were derived from the Australian Treasury which categorised costs at a national level into: (1) strict lockdown, (2) moderate lockdown, (3) low-level restrictions and (4) baseline (minimum public health) restrictions. Their associated costs were 3.2, 2.35, 0.65 and 0.1 billion AUD per week, respectively, at a national level. These costs do not include additional mental health or other costs associated with lockdowns [18].

**Table 4. Assumed Australian (NSW, VIC, and WA) vaccine effectiveness.**

| Susceptibility | Onward transmission | Hospitalisation rate | ICU rate | Ventilation rate | Mortality rate |
|---|---|---|---|---|---|
| 0.7 | 0.65 | 0.87 | 0.88 | 0.88 | 0.9 |

Source: Adapted from [4: Table S2.3 and S2.5].

We scaled the Australian Treasury costs to a state-level cost based on the Gross State Product as a proportion of Australia's Gross Domestic Product. We assumed that between a 70% and 80% vaccination rate at a state level when there was community transmission (NSW and VIC) that the lockdown costs range from low-level restrictions to moderate lockdown; at a 80% to 90% vaccination rate the public health measures cost were the equivalent of low-level restrictions; and with a 90% vaccination rate and above, baseline restrictions applied.

**2.2.5. Calibration process.** The system of finite-difference Eqs (1)–(8) was numerically solved [19,20] with the time step specified at one day to match the frequency of reported data. Time zero, the model start date, was specified as October 11[th] 2021. The maximum simulation time horizon was 4 years (1,460 days). The demand for health care services (hospitalizations, ICU, and ventilation) were calculated using Eqs (9)–(14). The rates of hospitalization, ICU, and ventilation in each state are reported in Table A-B in S2 Text. These state-specific rates were estimated, using the age distribution in Table 3, but scaled to match the aggregate values observed in actual data in Table 2.

# 3. Results

## 3.1. New South Wales (NSW)

The projected outcomes for NSW are presented in Fig 2, with numerical values summarized in Table A in S3 Text. Fig 2 summarizes six key public health outcomes for three scenarios: (i) lockdown ends when 70% of the eligible population is fully vaccinated (red colour), (ii) lockdown ends when 80% of the eligible population is fully vaccinated (yellow colour), and (iii) lockdown ends when 90% vaccination of the eligible population is fully vaccinated (green colour). The solid curves are the mean levels of the projected outcomes, and the bands represent their 95% confidence intervals. All six panels projected what would happen before and after a vaccination level (70%, 80% and 90%) were achieved, which are presented by the vertical lines.

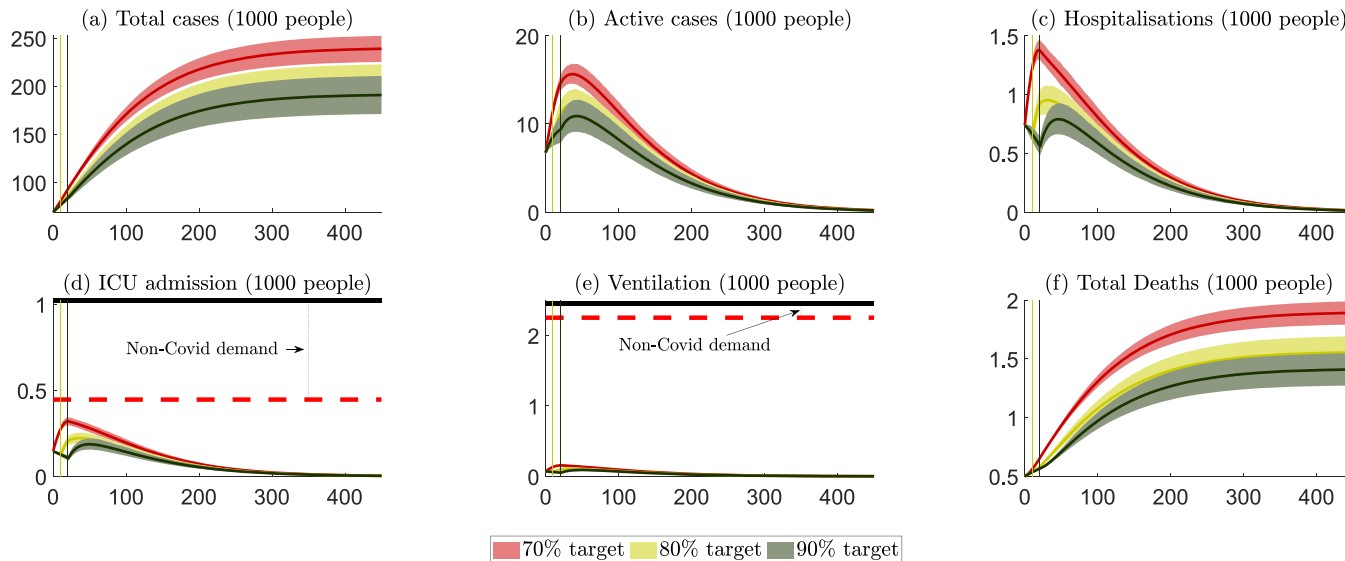

**Fig 2. NSW: Projected public health outcomes after opening up.** Notes about Fig 2: 1. Day zero is specified to be 11 October 2021. 2. The red, yellow, and green colours represent scenarios where lockdowns are removed after the 70%, 80% and 90% targets, respectively. 3. The (red, yellow, and green) vertical lines represent the days when the 70%, 80% and 90% targets are reached. 4. The (red, yellow, and green) curves represent the mean of all projection outcomes, and the bands represent their 95% confidence intervals. 5. The black horizontal lines in panels (d) and (e) are the capacity of the health care system, i.e., staffed ICU beds and ventilators (data sources reported in S1 Text). 6. The red horizontal dashed lines in panels (d) and (e) are the net capacity of the non-COVID19 ICU and ventilation demand, which are estimated via the occupation rates of ICU beds and ventilators in 2018/2019 (data sources reported in S1 Text).

On October 11[th] 2021, NSW achieved the 70% vaccination target. If the vaccination rate were maintained at the average of the 14-day period ending October 11[th] 2021 (i.e., around 64,000 people become fully vaccinated a day), it would take NSW around 10–11 days to achieve the 80% target and another 10–11 days to achieve the 90% target. These time periods are shorter than the vaccination progress for Australia to achieve the same vaccination rates.

Community transmission increases when lockdowns are relaxed. We highlight that the number of ICU beds and ventilators available for COVID-19 patients depends on the staffing capacity to maintain the high quality of care needed by patients in ICU. While there is an estimated 'surge' capacity for additional beds in ICU of about 800 for Australia, only half of this surge capacity could be staffed with suitably qualified and experienced medical personnel [21]. Thus, a capacity limit on suitably qualified and experienced staff places an upper limit on the net capacity of ICU beds and ventilators available for non-COVD-19 and COVID-19 patients. When patient demand is close to or exceeds this net capacity for ICU beds and ventilators, the fatality rate for non-COVD-19 and COVID-19 patients will likely rise. Table A in S4 Text compares the cost of health care services and patient welfare losses across different vaccination targets from opening up in NSW. The numbers in this table are costs beginning on October 11[th] 2021, and all costs incurred before that day, (including health service cost and welfare losses for recovered and non-recovered patients) are considered 'sunk' in that they would be incurred regardless of the opening-up decision. The costs of health care services would be some 8%-10% of the total costs in NSW. Opening up at a higher vaccination rate would reduce the total loss by 1.3 billion AUD should the vaccination rate be increased from 70% to 80%, and this would be further reduced by 0.5 billion AUD if lockdowns were maintained until the 90% vaccination rate were achieved.

Fig 3 provides three cost curves: (1) Health care costs and welfare losses from COVID-19; (2) Economy lockdown costs; and (3) Total costs, the sum of both health care costs and welfare losses and economy lockdown costs. The costs of health care services and welfare losses are *declining* in the vaccination rate before opening up because a greater level of vaccination means fewer hospitalizations and fatalities, all else equal. The economy lockdown costs are *increasing* in the vaccination rate as the higher is the vaccination rate at opening up the longer are the cumulative daily costs associated with a lockdown.

From a societal costs perspective, the minimum of the sum of the two costs (health care and welfare losses and economy lockdown costs) is the preferred vaccination rate for opening up noting a higher vaccination rate will always deliver lower health care costs and welfare losses but also higher lockdown costs. To the extent that vulnerable communities have a higher risk of contracting COVID-19 [22] and/or have a higher rate of hospitalisation or mortality, other than differences based on age, and also a *lower* vaccination rate than the state average, our estimates will underestimate the health care costs and welfare losses at the preferred vaccination rate.

The economy lockdown cost in NSW was estimated by scaling down the national economy cost of lockdowns per week (Table 5) using the share of NSW in the total GDP of Australia (31.7%) and the time required for NSW to reach a target for vaccination rate, i.e., 10–11 days to increase the vaccination rate from 70% to 80%. During this interval, the economy lockdown cost would be around 0.77 billion AUD. Fig 3 indicates that the preferred vaccination target for opening up in NSW is 80%.

We compared the three common vaccination targets for removing the lockdown in NSW with the preferred vaccination rate for NSW (80%) in Table A in S5 Text. Results in this table were converted to per-capita and percentage changes for comparison purposes. Opening up NSW at the 70% vaccination rate (with low-level to moderate public health restrictions in place) would incur additional health care and welfare losses of 158 AUD/person (about 28%

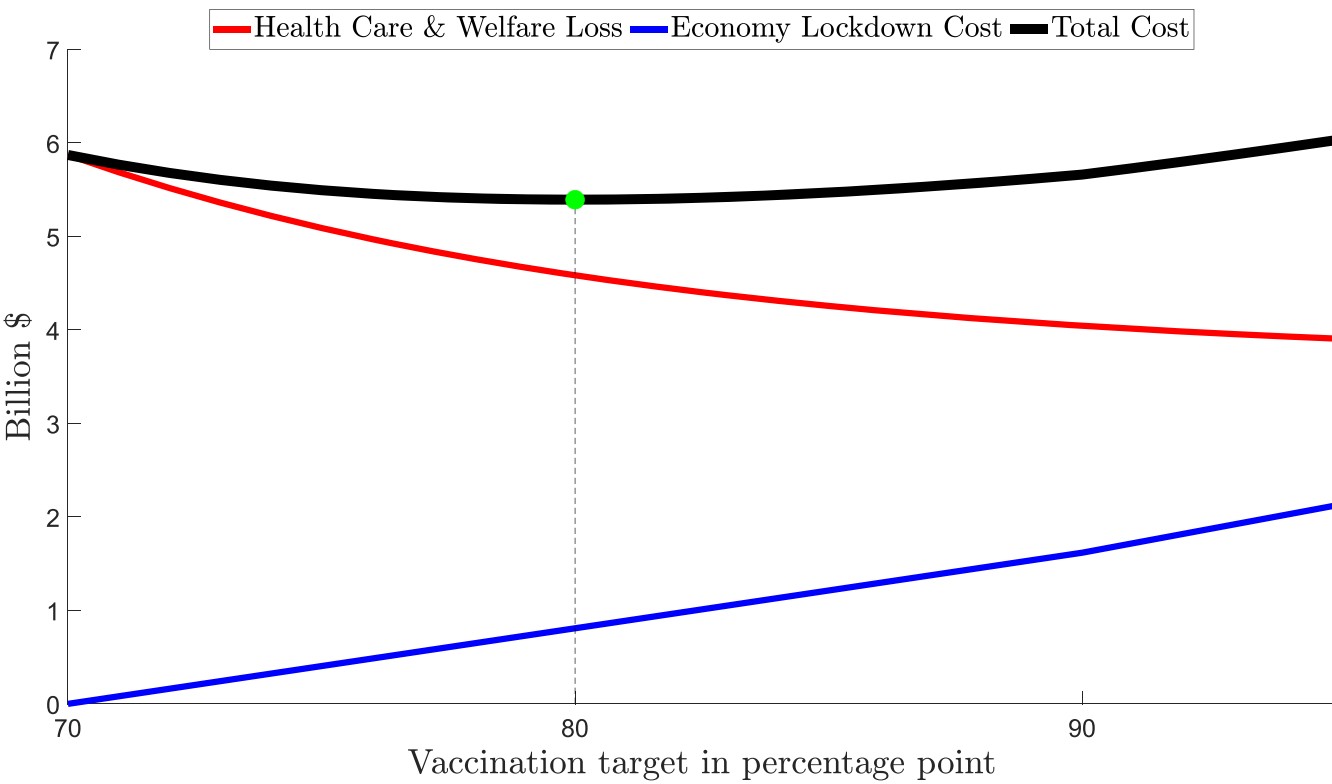

**Fig 3. NSW: Societal costs of different lockdown 'opening up' vaccination rates.** Notes about Fig 3: 1. All costs are counted from October 11th 2021. 2. Total costs = health care costs and welfare losses + economy lockdown costs.

higher than the cost at the economically justifiable target) and save approximately 99 AUD/person from *not* having a lockdown. Thus, the net societal loss from premature opening at a 70% vaccination rate, rather than at the preferred rate of 80%, is 59 AUD/person. Identical calculations are also provided in Table A in S5 Text for the 90% (33 AUD/person loss) vaccination rates noting the lowest total societal costs are at the preferred vaccination rate of 80%.

The sensitivity analyses of the preferred vaccination target are reported in Table B in S5 Text and Table C in S5 Text. The *less* effective are public health measures after opening up, the higher preferred vaccination target. This finding is consistent with [5] who used an agent-based modelling framework to show that, for Australia, the more effective are public health measures after opening up then the lower is the vaccination rate required to avoid adverse public health outcomes.

### 3.2. Victoria (VIC)

Public health outcomes for VIC are summarized in Fig 4, with numerical values summarised in Table B in S3 Text. If the vaccination rate is maintained at the average of the 14-day period ending October 11th 2021 (i.e., around 45,000 people become fully vaccinated per day), it would take VIC around 18, 30 and 43 days to achieve the 70%, 80%, and 90% rates, respectively. Our projected peak hospitalizations and ICU admissions are comparable to those of [4], respectively, at 1,200–2,500 and 260–550 under various scenarios from October to December 2021. As with NSW, opening up at a higher vaccination rate reduces cases, hospitalizations,

**Table 5. Australian Public Health Measures and Economy Costs by Stringency of Public Health Measures.**

|  | Strict lockdowns | Moderate lockdowns | Low level restrictions | Baseline restrictions |
|---|---|---|---|---|
| Economy cost | AUD 3.2 billion/week | AUD 2.35 billion/week | AUD 0.65 billion/week | AUD 0.1 billion/week |
| Stay at home orders | • Stay-at-home except essential purposes | • Stay-at-home except for work, study and essential purposes | • No stay-at-home orders | • No stay-at-home orders |
| Density restrictions | • 4 sqm rule | • 2 sqm rule | • 2 sqm rule | • 2 sqm rule |
| Retail trade | • Non-essential retailers and venues closed to public • Take away and home delivery only | • Increased retail activity, subject to density restrictions • Seated dining for small groups at cafes/restaurants | • Social distancing rules apply • Larger groups allowed | • Social distancing rules apply |
| Work | • Only workplaces categorised as permitted work allowed to operate on-site and subject to restrictions | • Work from home, if possible, capacity limits and restrictions on office space apply | • Return to work, but social distancing and capacity restrictions on office space apply | • 1.5 sqm rule |
| Schools and childcare | • Closed, remote learning only | • Closed or graduated return | • Open | • Open |
| Capacity restrictions | • No gatherings, non-essential venues etc. closed | • Indoor venues closed • Capacity limits restricted to small groups outdoors | • Recreational activities allowed and venues open but social distancing and capacity limits apply | • Large sporting venues to operate at 70% capacity |
| Travel restrictions | • Essential movements only within 5 or 10 km radius • No intra- or inter-state travel | • Non-essential travel limited; no intra or inter-state travel | • No travel restrictions • Interstate travel allowed | • No travel restrictions • Interstate travel allowed |
| • Other | • Curfew • No household visitors and 2-person limit on exercise | • 5 visitors to household and limited outdoor gatherings e.g., 10 people | • Requirements for record-keeping, COVID-safe plans |  |

Sources: Extracted from [8: Tables 4 and 5].

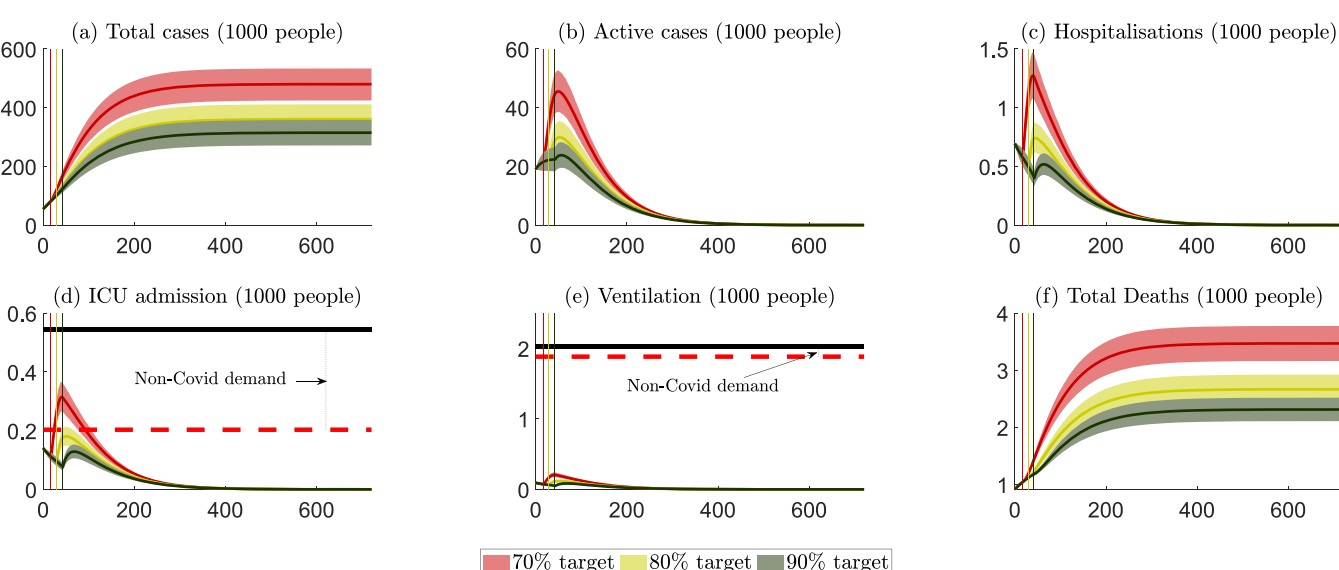

**Fig 4. VIC: Projected public health outcomes after opening up.** Notes about Fig 4: 1. Day zero is specified to be October 11[th] 2021. 2. The red, yellow, and green colours represent scenarios where lockdowns are removed after the 70%, 80% and 90% targets, respectively. 3. The (red, yellow, and green) vertical lines represent the days when the 70%, 80% and 90% targets are reached. 4. The (red, yellow, and green) curves represent the mean of all projection outcomes, and the bands represent their 95% confidence intervals. 5. The black horizontal lines in panels (d) and (e) are the capacity of the health care system, i.e., staffed ICU beds and ventilators (data sources reported in S1 Text). 6. The red horizontal dashed lines in panels (d) and (e) are the net capacity of the non-covid demand for ICU and ventilation, which are estimated via the occupation rates of ICU beds and ventilators in 2018/2019 (data sources reported in S1 Text).

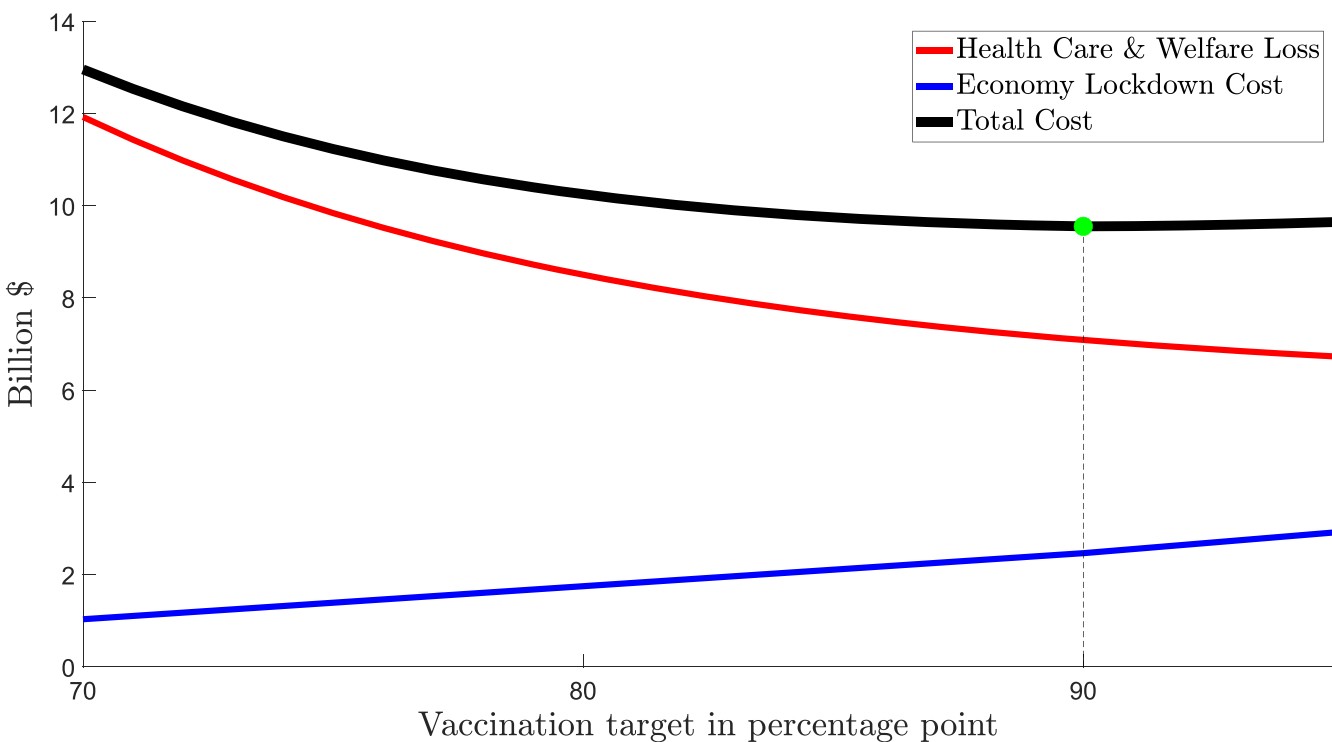

**Fig 5. VIC: Societal costs of different lockdown 'opening up' vaccination rates.** Notes about Fig 5: 1. All costs are counted from October 11[th] 2021. 2. Total costs = health care costs and welfare losses + economy lockdown costs.

ICU numbers and fatalities. Unlike NSW, VIC would exceed its maximum ICU capacity should it open up at a 70% vaccination rate.

Table B in S4 Text summarises the cost of health care services and patient welfare losses for VIC. Maintaining lockdowns while waiting for the vaccination rate to reach a higher level would reduce health care services costs and welfare losses. These avoided costs would be around 3.4 billion AUD if the lockdowns were to be extended from a 70% vaccination rate to the 80% vaccination rate, and would be 1.4 billion from the 80% target to the 90% target. Fig 5 summarizes the costs of different vaccination rates for VIC. It shows that the preferred vaccination rate that minimizes total societal costs in the state is 90%.

Table D in S5 Text compares vaccination rates for opening up in VIC. Results in this table are converted to per-capita and percentage changes for comparison purposes. Opening up VIC at the 70% vaccination rate (with low-level to moderate public health restrictions) would incur additional health care costs and welfare losses of approximately 723 AUD/person, more than three times the per capita cost savings from *not* having a lockdown of about 214 AUD/person, compared to the preferred vaccination rate of 90%. Thus, the net societal loss from premature opening at 70% vaccination rate, rather than at the preferred rate of 90%, is 508 AUD/person. Identical calculations are also provided in Table D in S5 Text for an 80% vaccination rate of about 104 AUD/person loss.

The sensitivity analyses of the preferred vaccination target are reported in Table E in S5 Text and Table F in S5 Text. The preferred vaccination rate is above or close to 90% in all considered scenarios. The total societal cost is increasing with lockdown costs and also the daily transmission during lockdown and is decreasing with the daily vaccination rate and also the reduction in community transmission after opening up.

### 3.3. Western Australia (WA)

In 2021 the COVID-19 epidemic in WA was very different to NSW and VIC. As of October 11[th] 2021, WA had no cases of community transmission with all active cases associated with arrivals from abroad or inter-state in supervised quarantine. WA is one of the most isolated jurisdictions in the world and almost all out-of-state arrivals come by air. This isolation, along with very strict protocols for entry and supervised quarantine, helped WA to maintain no community transmission throughout 2020 and 2021.

As of the model start date, WA did not have any significant restrictions within the state, i.e., people were requested to check-in when entering venues and follow some restrictions in relation to age care if they are symptomatic. Our estimates of the vaccination rate in WA to October 11[th] 2021 was 43%, with around 11,000 people becoming fully vaccinated per day. At this vaccination rate, it would have taken WA about 43, 64, and 84 days, respectively, to achieve the 70%, 80% and 90% vaccination rates.

In our model when WA opens its borders, some incoming passengers will seed the SARS-CoV2 virus in the community because many fully vaccinated cases are asymptomatic [23,24] and tests are not a 100% effective at identifying all positive cases. The epidemiology outcomes after opening up depend on how the state would respond if community transmission were to occur. Two important considerations are whether the state would close its border again (and if yes, when) and whether the state would impose public health restrictions should community transmission occur (and if yes, when).

We assumed that the state would not open its borders if it expected to close its border again in a short period of time. As the vaccination progress is foreseeable, we further assumed that the state would only choose to open early, before the vaccination progress is complete, if it would not reverse the policy, at least until all eligible and willing people are fully vaccinated. In addition, we assumed that if an outbreak continued 90 days after the vaccination progress was complete, the state would impose more restrictive health measures to reduce community transmission.

The projected public health outcomes should WA opens its borders at 70%, 80% and 90% vaccination rates, assuming that WA has no community transmission until it its state border fully opens up, are provided in Fig 6, with numerical values summarized in Table C in S3 Text. When the WA state border is opened in our model we assumed that interstate travel would return to its 2020/21 level (around 1,600 incoming passengers a day); all incoming passengers would be fully vaccinated, and 0.5% of the passengers would be asymptomatically infectious and undetected. We further assumed the effectiveness of the current (and minimal) public health restrictions in WA until opening up are able to reduce community transmission by 5%. Given that there is no available WA data for the Delta variant, we assume the hospitalization rate, ICU rate and mortality rate in the state when community transmission occurs was the average of NSW and VIC.

Fig 6 shows that COVID-19 outbreaks in WA would increase at a lower growth rate than projected for NSW and VIC. This is because when the state opens up in our modelling there is no community transmission, and the vaccination rate is at least 70% for those aged 12 years and older. By contrast to NSW and VIC, zero (or very low) COVID-19 in WA, allows for better epidemic control without lockdowns [25,26]. In all three scenarios, however, community transmission would continue after the vaccination progress is complete, and the state would need to apply more restrictive public health measures to control community transmission.

We assumed that WA's state border closure reduces the 'accommodation and food' sector valued at about $2 billion AUD per annum [27] by 50%, or equivalent to 40 million AUD per week. In Table C in S4 Text, if the WA border were opened at a 70% vaccination rate the

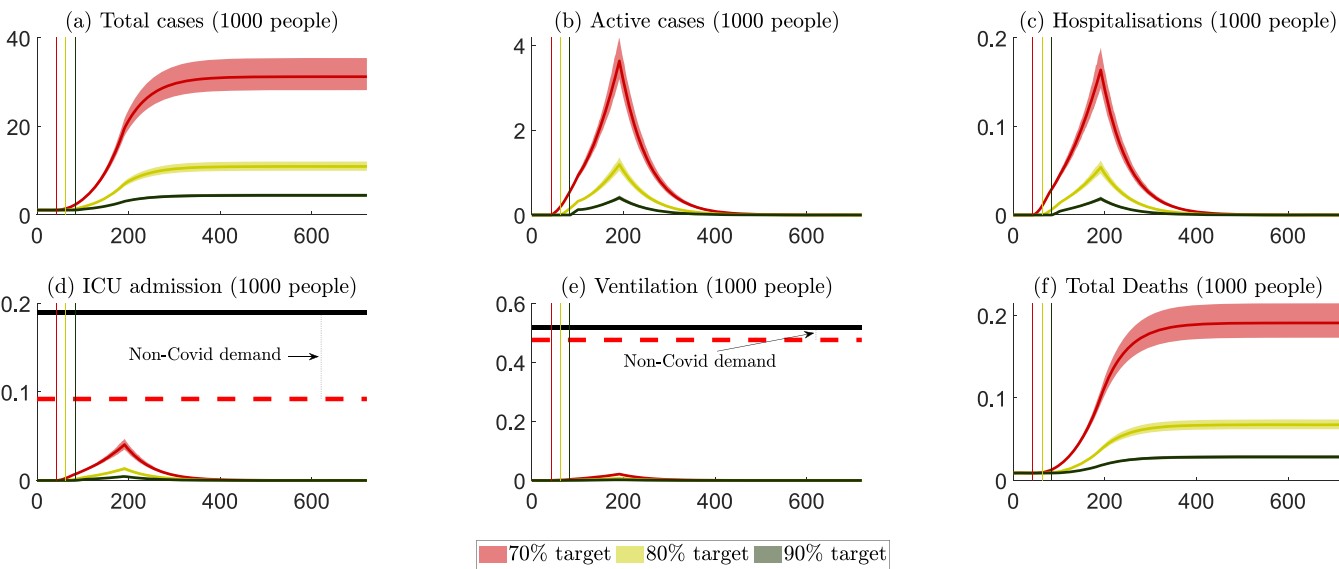

**Fig 6. WA: Projected public health outcomes after opening up.** Notes about Fig 6: 1. Day zero is specified to be October 11, 2021. 2. The red, yellow, and green colours represent scenarios where WA border is opened after the 70%, 80% and 90% targets, respectively. 3. The (red, yellow, and green) vertical lines represent the days when the 70%, 80% and 90% targets are reached. 4. The (red, yellow, and green) curves represent the mean of all projection outcomes, and the bands represent their 95% confidence intervals. 5. The black horizontal lines in panels (d) and (e) are the capacity of the health care system, i.e., staffed ICU beds and ventilators (data sources reported in S1 Text). 6. The red horizontal dashed lines in panels (d) and (e) are the net capacity of the non-covid demand for ICU and ventilation, which are estimated via the occupation rates of ICU beds and ventilators in 2018–2019 (data sources reported in S1 Text).

average health care costs and welfare losses would be 857 million AUD (95%CI ~ [763–971]). If WA's strict state border controls were maintained until it reached an 80% vaccination rate, the health care costs and welfare losses would be reduced to 278 million AUD (95%CI ~ [249–311]). If the state border controls were maintained until a 90% vaccination rate, these costs would further be reduced to 93 million AUD (95%CI~ [83–104]).

Fig 7 summarizes the projected costs of different vaccination rates for WA. It shows that the preferred vaccination rate that minimizes total societal costs in the state is 89%. If its state border remains closed until a higher vaccination rate were reached, the cost of health care services and welfare loss would be lower, but the border closure costs would be higher. At its vaccination progress (11,000 people becoming fully vaccinated per day) at the start of the model date, it would take WA approximately 43 days from October 11th 2021 to reach the 70% vaccination target, and the state border closure cost during this period would be some 246 million AUD.

Table G in S5 Text compares three vaccination rates (70%, 80% and 90%) with the preferred vaccination rate (89%) that minimizes societal costs for the state. Opening the state border at a 70% vaccination rate would incur additional health care costs and welfare losses of approximately 285 AUD/person, more than three-fold larger than the cost saving of 84 AUD/person from not having a state border closure, compared to the preferred vaccination rate (89%). Identical calculations are also provided in Table G in S5 Text for the 80% (26 AUD/person loss) and 90% (1 AUD/person loss) vaccination rates noting the lowest total societal costs are at the preferred vaccination rate of 89%.

Results of sensitivity analyses for WA show that the preferred vaccination rate is close to 90% in most scenarios (Table H in S5 Text and Table I in S5 Text). In terms of the probability of COVID-19 seeding from out-of-state arrivals, a much higher proportion of fully vaccinated arrivals could be asymptomatic [23,24,28]. Thus, our baseline assumption that 0.5% of fully vaccinated arrivals are asymptomatic and undetected after opening up of the state border is, in

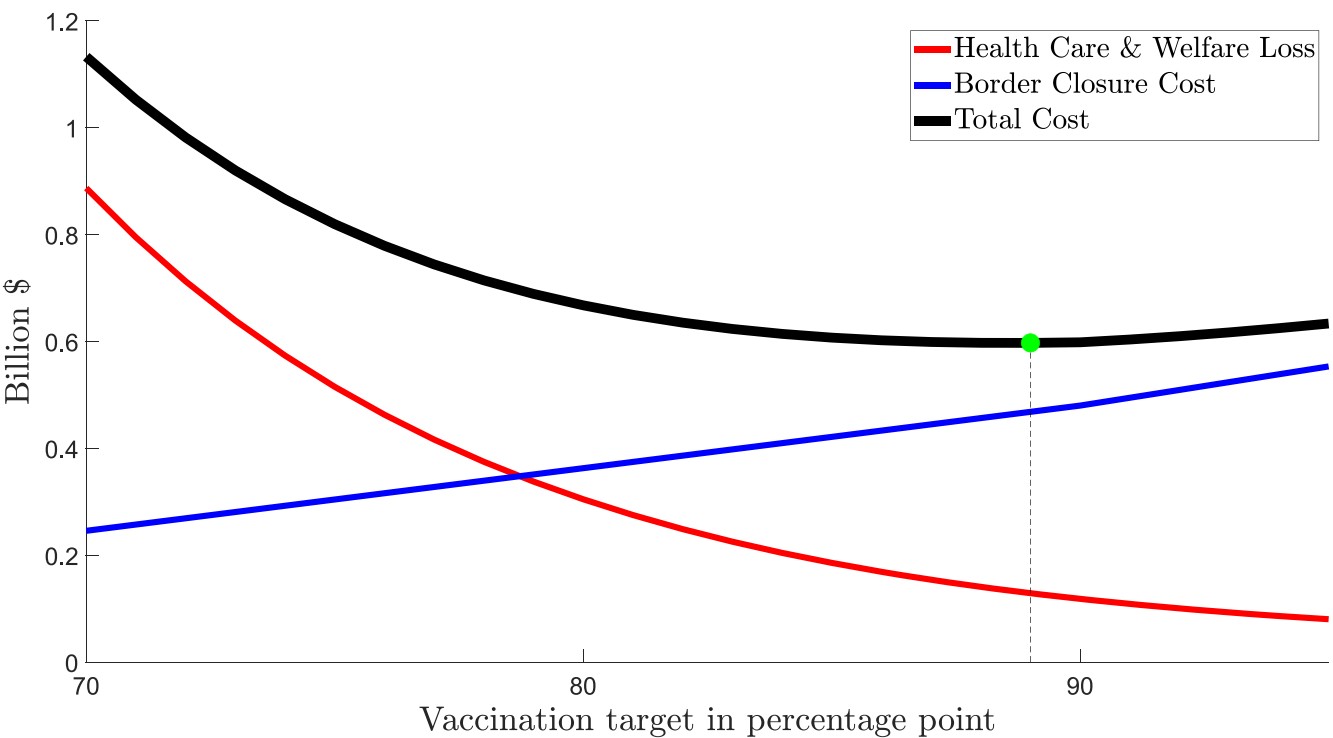

**Fig 7. WA: Societal costs of different vaccination rates with respect to state border reopening.** Notes about Fig 7: Including costs that vary with vaccination targets, counted from October 11[th] 2021. Costs appliable to ALL vaccination targets are not included (e.g., vaccination costs, the cost of maintaining the minimal restrictions, the cost of policy interventions after the vaccination target is achieved).

our view, a reasonable assumption. We highlight that the higher is the seeding rate of COVID-19 from arrivals, the higher is the preferred vaccination rate.

## 4. Discussion

Our results provide important insights for decision-makers when determining the vaccination rate that minimizes total societal costs from opening up an economy that is in lockdown to reduce community transmission of COVID-19 or has imposed strict border controls. First, our projected public health outcomes are comparable, adjusted for differences between state and the national population, to the high seeding from arrivals [29, p:8–10], with baseline public health and safety measures and partially effective testing, tracing, isolation and quarantine measures scenario. Further, our results are consistent with both the Doherty Institute and the Grattan Institute [7] that also find that the public health outcomes from opening up at a 80% vaccination rate are larger than at a 70% vaccination rate.

Second, we provided a method to calculate the preferred vaccination rate to open up a locked down economy with community transmission (NSW and VIC) and showed that this preferred vaccination rate depends on several key factors. We found that the preferred population vaccination rate is increasing: (1) the lower is the daily lockdown cost; (2) the larger are the public health costs from COVID-19; (3) the higher is the rate of community transmission before opening up; and (4) the less effective are the public health measures after opening up (see Fig 8).

Third, in the case of a state with no community transmission (WA), we found the preferred vaccination rate is increasing: (1) the less effective are the public health measures after opening

**Fig 8. Summary of the sensitivity of the economically preferred vaccination target.**

up; (2) the lower are border closure costs; and (3) the greater is the probability of COVID-19 seeding into the community from out-of-state arrivals.

Fourth, the costs of opening up too soon are asymmetric such that the losses from opening prematurely are greater than the losses from opening too late for the same percentile error difference in the preferred vaccination rate. In the context of Australia, we also found that opening up prematurely, for example at a 70% vaccination rate, imposes substantial per capita costs estimated at 59 AUD (NSW), 508 AUD (VIC) and 201 AUD (WA). Importantly, we projected that the combined demand for staffed ICU beds by COVID-19 and non-COVID-19 patients would likely exceed the state-level current and surge capacity at 70% vaccination rate for both NSW and VIC, and mights also exceed the state-level capacity of VIC at an 80% vaccination rate.

Fifth, our estimated preferred vaccination rates for NSW (80%), VIC (90%) and WA (89%) all exceeded the Phase C vaccination targets (16 years and older) under Australia's National Plan to transition the response to COVID-19 [2]. This finding is consistent with the Grattan Institute's recommendation, based on projected public health outcomes, to fully vaccinate 80% of all eligible Australians (and 95% of those 70 years and older) before opening up the international borders and no longer using lockdowns [7].

Under Australia's National Plan, Phase B commences when 70% of those 16 years and older are fully vaccinated. Phase B coincides with eased restrictions on vaccinated persons and has as a key goal to minimise on-going community cases with low-level restrictions. Phase C of the National Plan commences when 80% of those 16 years and older are fully vaccinated. Under Phase C, fully vaccinated people would be exempt from travel restrictions and a key goal would be to have only highly targeted lockdowns. The differences between our estimated preferred vaccination rates (NSW 80%, VIC 90%, and WA 89%) and the 70% and 80% targets in the National Plan are more pronounced than they appear because a 70% and 80% vaccination of those 12 years and older is 'equivalent' to a 74% and 85% vaccination rate of those 16 years and older. Our preferred state-level vaccination rates would be higher again should there be vulnerable populations, with high probabilities of morbidity and mortality from COVID-19 independent of age, and if these sub-populations were vaccinated at less than the preferred state-level average vaccination rate.

While our model and results were developed for three Australian states there were, nevertheless, of *ex-ante* relevance to other jurisdictions in Australia, such as Tasmania and the Northern Territory, and elsewhere, such as New Zealand [30], which at the time of modelling in October 2021 had yet to either open up their respective borders and/or open up their locked down economies. Our approach to estimating total societal costs and the preferred vaccination rate also allows for an *ex-post* assessment of decisions of other countries that have already opened up. For instance, the United Kingdom opened up on the July 19th 2021 when 67% of its population 16 years and older was vaccinated while Denmark opened up on September 10th 2021 when 80% of its population 16 years and older was vaccinated [31].

Our model could be extended, if daily data were available, to incorporate partial vaccination and reinfection and to accommodate three connected SIRM components: unvaccinated, partially vaccinated, and fully vaccinated for each of our ten age groups. Unfortunately, daily data for each age group at each step of the vaccination rollout, noting the different vaccines used had different dose intervals, are not available for Australia. Consequently, our model does not control for waning vaccine effectiveness (before a booster dose).

We also highlight that in our modelling vaccination is for the susceptible category only. If the data were available, our model could identify the vaccination rates of people who became vaccinated after recovering from COVID-19 and also the effectiveness of post-recovery vaccination. Further, our model could be extended to include post-recovery vaccination and

reinfection by adding flows from recovery to susceptible compartments. We also note that while all 12+people were eligible for vaccination on 11 October 2021, there were substantially different rates of vaccination by age groups. This is because Australian governments had priortized older cohorts for vaccination before younger people. If we had realiable daily time-series data on the vaccination rate by each age group from the beginning of vaccination program in February 2021, our model could have been calibrated to simulate epidemiological and economic outcomes under alternative priortization strategies, and also to evaluate the impact of age priorization with respect to vaccination.

## 5. Conclusions

As countries transition in their responses to the COVID-19 pandemic through vaccinations, a fundamental question is: What is the preferred vaccination rate that minimises societal costs for opening up public health measures or state border controls? Using a separate age-structured SIRM model for three jurisdictions in Australia (New South Wales, Victoria and Western Australia), we estimated a preferred vaccination rate for opening up, by state, that minimizes the sum of health care costs, welfare losses from fatalities and those recovering from COVID-19, and economy lockdown costs and/or state border control costs.

Our results showed that the target vaccination rates under Australia's National Plan to transition its COVID-19 response are *lower* than the vaccination rate we estimate that would minimize state-level societal costs. We also found that opening up at a lower than preferred vaccination rate, such as 70% under Phase B of the National Plan, would impose substantial per capita societal costs. In the states of New South Wales and Victoria, opening up at a 70% vaccination rate the projected ICU patient demand would be expected to exceed the available state-level staffed ICU bed capacity.

The methods we used to estimate a preferred vaccination rate can be applied to any jurisdiction where there are available data. Our results also provide useful guidance as to the qualitative effects of different factors, such as; the speed of vaccination, the effectiveness of low-level public health measures, among others, on the preferred vaccination rate that minimises total societal costs. In sum, our findings provide valuable insights for decision makers to determine the cost-minimizing vaccination rate to open up an economy in lockdown and/or with strict border controls on arrivals.

## Supporting information

**S1 Text. Data sources.**
(DOCX)

**S2 Text. Data-matched age distributions.** Table A NSW: Hospitalization, ICU admissions, ventilation, and fatality. Table B Victoria: Hospitalization, ICU admissions, ventilation, and fatality. Table C Western Australia (the average of NSW and Victoria).
(DOCX)

**S3 Text. Projected epidemiological outcomes.** Table A NSW: Projected outcomes at different vaccination rates for opening up (1000's people). Table B VIC: Projected outcomes at different vaccination rates for opening (1000's people). Table B WA: Projected outcomes at different vaccination rates for opening up.
(DOCX)

**S4 Text. Estimates of health care and welfare losses at different vaccination rates.** Table A NSW: Estimates of health care and welfare losses, billions AUD. Table B VIC: Estimates of health care and welfare losses, billions AUD. Table C WA: Projected health care costs and

welfare losses, millions AUD.
(DOCX)

**S5 Text. Sensitivity analysis.** Table A NSW: Per capita (AUD and %) differences from Opening Up at alternative vaccination rate to the preferred vaccination rate (80%). Table B NSW: Sensitivity of preferred vaccination rate to the effectiveness of public health measures after opening up at reducing community transmission and the speed of vaccination. Table C NSW: Sensitivity of preferred vaccination rate to economy lockdown cost and community transmission during lockdown. Table D VIC: Per capita (AUD and %) cost differences of alternative vaccination rates to the preferred vaccination rate (90%). Table E VIC: Sensitivity of the preferred vaccination rate to the effectiveness of public health measures after opening up and speed of vaccination. Table F VIC: Sensitivity of preferred vaccination rate to economy lockdown costs and rate of community transmission during lockdown Table G WA: Per capita (AUD and %) cost differences of alternative vaccination rates to the preferred vaccination rate (89%). Table H WA: Sensitivity of preferred vaccination rate to the effectiveness of public health measures after opening up at reducing community transmission and speed of vaccination. Table I WA: Sensitivity of preferred vaccination rate to opening up with the probability of seeding and state border closure costs.
(DOCX)

## Author Contributions

**Conceptualization:** R. Quentin Grafton, Tom Kompas.

**Data curation:** Long Chu, R. Quentin Grafton, Tom Kompas.

**Formal analysis:** Long Chu, R. Quentin Grafton, Tom Kompas.

**Investigation:** R. Quentin Grafton, Tom Kompas.

**Methodology:** Long Chu, R. Quentin Grafton, Tom Kompas.

**Resources:** Long Chu, R. Quentin Grafton, Tom Kompas.

**Software:** Long Chu, Tom Kompas.

**Validation:** Long Chu, R. Quentin Grafton, Tom Kompas.

**Visualization:** Long Chu, R. Quentin Grafton.

**Writing – original draft:** Long Chu, R. Quentin Grafton, Tom Kompas.

**Writing – review & editing:** Long Chu, R. Quentin Grafton, Tom Kompas.

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
