## [Decision Letter · Decision Letter 0]

6 Dec 2021

PGPH-D-21-00834

What vaccination rate(s) minimise total societal costs after 'opening up' to COVID-19? Age-structured SIRM Results for the Delta variant in Australia (New South Wales, Victoria and Western Australia)

Dear Dr. Kompas,

Thank you for submitting your manuscript to PLOS Global Public Health. After careful consideration, we feel that it has merit but does not fully meet PLOS Global Public Health’s publication criteria as it currently stands. Therefore, we invite you to submit a revised version of the manuscript that addresses the points raised during the review process.

Thank you for your submission on this timely and important issue. As you will see, the reviewers have identified some substantial concerns with your methods, which I think you can potentially address with a major revision. I encourage you to revise and resubmit this work to PLOS Global Public Health.

We look forward to receiving your revised manuscript.

Kind regards,

Brooke E. Nichols

Academic Editor

Journal Requirements:

1. We ask that a manuscript source file is provided at Revision. Please upload your manuscript file as a .doc, .docx, .rtf or .tex. If you are providing a .tex file, please upload it under the item type ‘LaTeX Source File’ and leave your .pdf version as the item type ‘Manuscript’.

2. Please provide separate figure files in .tif or .eps format only, and remove any figures embedded in your manuscript file.  If you are using LaTeX, you do not need to remove embedded figures.

For more information about figure files please see our guidelines: https://journals.plos.org/globalpublichealth/s/figures

3. Your manuscript is missing the following sections: Methods and Results Section. Please ensure these are present, and in the correct order, and that any references to subheadings in your main text are correct. An outline of the required sections can be consulted in our submission guidelines here: https://journals.plos.org/globalpublichealth/s/submission-guidelines

4. Please update the completed 'Competing Interests' statement, including any COIs declared by your co-authors. If you have no competing interests to declare, please state "The authors have declared that no competing interests exist". Otherwise please declare all competing interests beginning with the statement "I have read the journal's policy and the authors of this manuscript have the following competing interests:"

Additional Editor Comments (if provided):

Reviewers' comments:

Reviewer's Responses to Questions

**Comments to the Author**

1. Does this manuscript meet PLOS Global Public Health’s publication criteria? Is the manuscript technically sound, and do the data support the conclusions? The manuscript must describe methodologically and ethically rigorous research with conclusions that are appropriately drawn based on the data presented.

Reviewer #1: Yes

Reviewer #2: Yes

2. Has the statistical analysis been performed appropriately and rigorously?

Reviewer #1: Yes

Reviewer #2: I don't know

3. Have the authors made all data underlying the findings in their manuscript fully available (please refer to the Data Availability Statement at the start of the manuscript PDF file)?

Reviewer #1: Yes

Reviewer #2: Yes

4. Is the manuscript presented in an intelligible fashion and written in standard English?

Reviewer #1: Yes

Reviewer #2: Yes

5. Review Comments to the Author

Reviewer #1: In this study, the authors use an age-structured compartmental model to estimate the health and economic impacts of ending COVID-19 control measures at different levels of vaccine uptake. The study is well presented, with methods and results explained in detail, and I credit the authors for releasing their data and code.

My major critique of the paper is that while the authors use an age-structured model, there is no investigation of the effect of vaccinating different proportions of different age groups and this is not mentioned as a limitation in the discussion. As age is the largest risk factor for severe COVID outcomes, the relative burden of, for example, 10% remaining unvaccinated 65+ is substantially greater than 10% remaining unvaccinated 18-65s. I believe this should be investigated and should be relatively straightforward to implement with your model.

Additional comments:

The discussion section is lacking in explanation of model limitations.

The authors only consider double vaccinated, which may underestimate vaccine effects from single doses.

No movement back to susceptible from vaccinated or recovered (i.e no waning) - this appears to be a significant factor in countries which achieved high vaccination rates early e.g UK, Israel.

Reviewer #2: In this manuscript, Chu and colleagues present model-based estimates of the optimal COVID-19 vaccination rates needed in three Australian states to minimize societal costs before lifting restrictions. They find that the optimal vaccination rate exceeds 80% in all three states, indicating that it may be a cost-effective strategy to extend costly restrictions until vaccination rates are high.

Overall, I found the analysis to be thorough and appropriate. I especially appreciate the range of sensitivity analyses that the authors included. The authors do a good job of situating their findings in the global context, noting that the experience with COVID-19 in Australia is unusual and yet offers valuable insights that may apply more broadly.

I have two main criticisms of the manuscript. First, the treatment of vaccination in the model wasn't fully clear to me. I may have missed it (apologies if so!), but I can't find a definition for the V_i parameter, which is presumably the daily vaccination rate, in Equation (1). I also worry about treating V_i as a constant, since this could lead to non-sensical model solutions (e.g. vaccinating > 100% of the population). Long after the model has been defined, the authors say that they reduce the vaccination rate by 20% once 90% vaccination rates have been achieved; this seems arbitrary and doesn't fix the misspecification issues. I elaborate on these issues further below in the minor comments.

The second main criticism is that I found the manuscript so long and dense that it took a great deal of effort to pull out the most salient points. The authors have useful things to say here, and I worry that those pearls will get drowned out by the surrounding text. I strongly advocate for the authors to put more of the figures and tables and associated text into a supplement, leaving the main text to describe just the central findings and their importance.

Minor comments:

70: The term "zero COVID" felt unnecessary and out of place here. I think of "zero COVID" as a policy aim, rather than an epidemiological phenomenon. I recommend simply cutting this.

Table 1: Could you include the overall totals for each state as well? This would be helpful for interpreting the case counts in your simulation figures.

128-134: V_i should also be defined in this paragraph.

Equation 1: a few comments here. First, (here and in all equations), for ordinary differential equations, it's more customary to use a common lower-case 'd' rather than the curly 'd' which usually represents partial derivatives. Also, I'd recommend using a different letter (not 'i', which is used in the numerator) to index the sums in the denominator (also in Equations 2, 5, and 6).

Next: some questions about the treatment of vaccination. As it stands, if V_i is a constant, you'll eventually reach some time where you've vaccinated >100% of the population. How did you account for this? Was there some stopping rule to ensure that you didn't end up with non-sensical simulations? Rather than V_i, you might be able to use a term like - Exp[-V_i t], which with the proper initial conditions will solve out to be a curve that (a) starts at 0 with a slope of V_i and (b) saturates at N as time -> infinity.

Also: why is the vaccination term exclusively in the susceptible category? Presumably people who are recovered can also get vaccinated. If you end up with an epidemic that infects >10% of the population before you reach 90% vaccination rates (seems possible), then as the model stands, you'll never reach 90% vaccination rates. This possibility isn't discussed, and I'd like to know how you accounted for it.

143: "fixed reduction in community transmission associated with opening up" - this doesn't make sense to me; wouldn't community transmission increase with opening up?

208-210: the claim that the "numbers of people being fully vaccinated... are proportional to the number of people waiting for vaccinations" doesn't seem to align with the model equations. For this to be true, I'd expect the vaccination rate to be multiplied by the compartment size.

251: I think there's a cross-referencing issue here; the equation are 1-8.

252: Is this time step for the simulation output or for the solving algorithm? If it's for the algorithm, then I worry that the step may be too large. One-day steps could lead to some large numerical errors. How did you check for such errors?

263-265: Again, confused by the treatment of vaccination. How was this reduction in vaccination rate implemented in the model?

Fig 2: the caption didn't make sense to me. It seems to me that this is just the number of vaccinated individuals over time.

275: How did you calculate confidence intervals from a deterministic ODE model?

Fig 3 would benefit from a legend describing the colors within the figure itself.

6. PLOS authors have the option to publish the peer review history of their article (what does this mean?). If published, this will include your full peer review and any attached files.

**Do you want your identity to be public for this peer review?** For information about this choice, including consent withdrawal, please see our Privacy Policy.

Reviewer #1: **Yes: **Billy J Quilty

Reviewer #2: No

---

## [Decision Letter · Decision Letter 1]

15 Mar 2022

PGPH-D-21-00834R1

What vaccination rate(s) minimize total societal costs after 'opening up' to COVID-19? Age-structured SIRM Results for the Delta variant in Australia (New South Wales, Victoria and Western Australia)

Dear Dr. Kompas,

Thank you for submitting your manuscript to PLOS Global Public Health. After careful consideration, we feel that it has merit but does not fully meet PLOS Global Public Health’s publication criteria as it currently stands. Therefore, we invite you to submit a revised version of the manuscript that addresses the points raised during the review process.

We look forward to receiving your revised manuscript.

Kind regards,

Brooke E. Nichols

Academic Editor

Journal Requirements:

1. Please amend your Financial Disclosure statement. If you did not receive any funding for this study, please simply state: “The authors received no specific funding for this work.”

2. Please update your Competing Interests statement. If you have no competing interests to declare, please state: “The authors have declared that no competing interests exist.”

Additional Editor Comments (if provided):

Reviewers' comments:

Reviewer's Responses to Questions

**Comments to the Author**

1. If the authors have adequately addressed your comments raised in a previous round of review and you feel that this manuscript is now acceptable for publication, you may indicate that here to bypass the “Comments to the Author” section, enter your conflict of interest statement in the “Confidential to Editor” section, and submit your "Accept" recommendation.

Reviewer #3: (No Response)

2. Does this manuscript meet PLOS Global Public Health’s publication criteria? Is the manuscript technically sound, and do the data support the conclusions? The manuscript must describe methodologically and ethically rigorous research with conclusions that are appropriately drawn based on the data presented.

Reviewer #3: Yes

3. Has the statistical analysis been performed appropriately and rigorously?

Reviewer #3: Yes

4. Have the authors made all data underlying the findings in their manuscript fully available (please refer to the Data Availability Statement at the start of the manuscript PDF file)?

Reviewer #3: Yes

5. Is the manuscript presented in an intelligible fashion and written in standard English?

Reviewer #3: Yes

6. Review Comments to the Author

Reviewer #3: (No Response)

7. PLOS authors have the option to publish the peer review history of their article (what does this mean?). If published, this will include your full peer review and any attached files.

**Do you want your identity to be public for this peer review?** For information about this choice, including consent withdrawal, please see our Privacy Policy.

Reviewer #3: No

---

## [Editor Report · Decision Letter 2]

27 Apr 2022

What vaccination rate(s) minimize total societal costs after 'opening up' to COVID-19? Age-structured SIRM Results for the Delta variant in Australia (New South Wales, Victoria and Western Australia)

PGPH-D-21-00834R2

Dear Prof Kompas,

We are pleased to inform you that your manuscript 'What vaccination rate(s) minimize total societal costs after 'opening up' to COVID-19? Age-structured SIRM Results for the Delta variant in Australia (New South Wales, Victoria and Western Australia)' has been provisionally accepted for publication in PLOS Global Public Health.

Best regards,

Brooke E. Nichols

Academic Editor